# Sporopollenin Capsules as Biomimetic Templates for the Synthesis of Hydroxyapatite and β-TCP

**DOI:** 10.3390/biomimetics9030159

**Published:** 2024-03-04

**Authors:** Arianna De Mori, Daniel Quizon, Hannah Dalton, Berzah Yavuzyegit, Guido Cerri, Milan Antonijevic, Marta Roldo

**Affiliations:** 1School of Pharmacy and Biomedical Sciences, University of Portsmouth, St Michael’s Building, White Swan Road, Portsmouth PO1 2DT, UK; arianna.demori@port.ac.uk (A.D.M.); up934323@myport.ac.uk (D.Q.); hannahrosedalt@gmail.com (H.D.); berzah.yavuzyegit@port.ac.uk (B.Y.); 2Mechanical Engineering Department, Recep Tayyip Erdogan University, Rize 53100, Turkey; 3Department of Architecture, Design and Urban Planning, GeoMaterials Laboratory, University of Sassari, 07100 Sassari, Italy; gcerri@uniss.it; 4School of Chemistry and Chemical Engineering, Faculty of Engineering and Physical Sciences, University of Surrey, Guildford GU2 7XP, UK; m.antonijevic@surrey.ac.uk

**Keywords:** sporopollenin, bio-template, calcium phosphate, hydroxyapatite

## Abstract

Pollen grains, with their resilient sporopollenin exine and defined morphologies, have been explored as bio-templates for the synthesis of calcium phosphate minerals, particularly hydroxyapatite (HAp) and β-tricalcium phosphate (TCP). Various pollen morphologies from different plant species (black alder, dandelion, lamb’s quarters, ragweed, and stargazer lily) were evaluated. Pollen grains underwent acid washing to remove allergenic material and facilitate subsequent calcification. Ragweed and lamb’s quarter pollen grains were chosen as templates for calcium phosphate salts deposition due to their distinct morphologies. The calcification process yielded well-defined spherical hollow particles. The washing step, intended to reduce the protein content, did not significantly affect the final product; thus, justifying the removal of this low-yield step from the synthesis process. Characterisation techniques, including X-ray diffraction, scanning electron microscopy, Fourier-transform infrared spectroscopy, and thermal gravimetric analysis, confirmed the successful calcification of pollen-derived materials, revealing that calcified grains were principally composed of calcium deficient HAp. After calcination, biphasic calcium phosphate composed of HAp and TPC was obtained. This study demonstrated the feasibility of using pollen grains as green and sustainable bio-templates for synthesizing biomaterials with controlled morphology, showcasing their potential in biomedical applications such as drug delivery and bone regeneration.

## 1. Introduction

Pollen grains play an important role in the reproduction of seed plants by protecting the genetic material [1,2]. They exhibit a highly resilient exine, or external shell, made of sporopollenin that protects the intine and pollen from environmental injury caused by microbial damage, extreme temperatures, and dehydration [3]. The intine, or internal shell, holds the male gamete and consists of pectin and cellulose [4,5]. Sporopollenin is composed of phenylpropanoids, phenols, and fatty acids and it is extremely resistant to pH and temperature [6,7]; this makes pollen grains able to withstand harsh pH conditions, such as those encountered in the stomach, making them attractive for the development of oral drug and vaccine delivery systems [5,6].

Pollen grains possess a wide range of different morphologies, sizes, and surface patterns [8]. A variation in morphology occurs between entomophilous (pollinated by insects) and anemophilous (wind pollinated) [9]. Elaborate morphologies in entomophilous plant pollens are important for pollinators attachment and adherence to the stigma surface; these pollens are protein-rich, heavy, and sticky in order to aid this process [10]. Smoother morphologies may be found in pollen from anemophilous grass and tree species. These pollens may also possess air sacs and will often be extremely lightweight to facilitate the wind pollination [9]. These different morphologies could be exploited as bio-templates for the production of various biomaterials with defined and hierarchical architecture in the same way in which the natural process of bio-mineralisation uses complex organic templates to create minerals with multiscale architecture [11]. Based on this idea, Zheng et al. used pollen grains as bio-templates for the production of bioactive glass particles obtaining a combined macro- and nano-porous structure [12]. This architecture allowed for the loading and controlled release of a model drug as well as for calcification when immersed in simulated body fluid, supporting the potential use of such structure both for drug delivery and for bone regeneration applications. Further uses of pollen grains as bio-templates have been recently reviewed by Wang et al. [11]; however, research in the use of pollen grains as bio-templates for the synthesis of materials for bone regeneration is extremely scarce.

In the present study, different pollen grain morphologies have been taken into consideration in order to study their effect on the synthesis of calcium phosphate minerals. The grains selected were from black alder (*Alnus glutinosa*), which is oblate; dandelion (*Taraxacum vulgare*), which is spheroidal, spiky, and possesses long spines; lamb’s quarters (*Chenopodium album*), a spheroidal and periporate pollen; ragweed (*Ambrosia elatior*), which is spheroidal with spines; and finally, stargazer lily (*Lilium orientalis*), which presents monosulcate pollen grains with a reticulate exine sculpturing [13]. In order for pollen grains to be used by humans, the intine contents must be removed to avoid any adverse effects caused by allergenic material, and the apertures in the pollen walls must be opened to create pores [6]. These apertures are useful because they can be used to fill the sporopollenin microcage with drugs or proteins, and create further morphological features to enhance the hierarchical structure [6]. Pollens can be cleaned through acidolysis and base hydrolysis; however, this method has been shown not to be successful in pollens such as ragweed (*Ambrosia artemisiifolia*) and sunflower (*Helianthus annuus*) due to the formation of an irreversible insoluble cake from which the pollen grains cannot be salvaged [4,5]. This limitation has led to many other methods being developed, including acidolysis without base hydrolysis, the acidolysis and base hydrolysis steps being switched, and other methods such as using ionic liquids [14].

The use of natural sophisticated structures as bio-templates provides a simple and reproducible way of synthesizing biomaterials of controlled morphology without using complex methodologies [15]. The scope of the present work was that of investigating if pollen grains can be used as bio-templates in the synthesis of hydroxyapatite (Ca_10_(PO_4_)(OH)_2_) and other calcium phosphate salts to obtain well-defined spherical hollow particles.

## 2. Materials and Methods

### 2.1. Materials

Ragweed (*Ambrosia elatior*, #AMBE.0115, dia: 16–22 µm), alder (*Alnus glutinosa*, #ALNG.0218, dia: 10–30 µm), dandelion (*Taraxacum vulgare*, #TARV.0115, defatted by acetone, dia: 25–46 µm), and lamb’s quarter (*Chenopodium album*, #CHEA.0215, defatted by acetone, dia: 22–31 µm) pollen grains were obtained from Pharmallerga (Lisov, Czech Republic). Stargazer lilies (*Lilium orientalis* ‘Stargazer’) were locally purchased, the flowers were allowed to bloom, and the pollen was collected over a period of few days. Ammonia solution 35%, hydrochloric acid 37%, and Phosphate Buffered Saline (PBS) were purchased from Fischer Scientific (Loughborough, UK). A phosphoric acid 85% solution in water, acetone 99.6%, and calcium chloride dihydrate +99% were all obtained from Acros Organics (Geel, Belgium). Sodium phosphate monobasic monohydrate and a QuantiPro BCA protein assay kit were purchased from Sigma-Aldrich (Poole, UK). Ethanol absolute was purchased from VWR chemicals (Leicester, UK).

### 2.2. Purification of Exine Capsules from Raw Pollen Grains

In total, one gram of each pollen was suspended in 15 mL of 85% *w*/*w* phosphoric acid and the mixture was stirred at 540 rpm at 70 °C for 5 h [6,11]. The product was collected by vacuum filtration through a fibre glass filter (MF200, Fisher Scientific, Loughborough, UK) with washings as follows: deionized water (500 mL), acetone (200 mL), 2 M HCl (100 mL), water (500 mL), acetone (100 mL), ethanol (200 mL), and water (100 mL); all solvents were heated on a hot plate and used hot. The exine capsules were then placed in a Petri dish and dried in a vacuum oven (Gallenkamp, UK). They were then stored in a desiccator until further use.

### 2.3. Bicinchoninic Acid (BCA) Protein Assay

Pollen protein extracts were produced by suspending 25 mg of each pollen in 6 mL of PBS for 4 h with continuous stirring at 25 °C. This was followed by centrifugation at 2000 rpm for 5 min (Jouan B4i Multifunction Centrifuge Series, Thermo Fisher Scientific, MA, USA) [16]. The supernatant was analysed using the QuantiPro BCA protein assay kit, as per manufacturer’s instructions. Supernatant from the washed pollen samples was used as collected, while supernatant from the unwashed samples was diluted in PBS. After a 1 h incubation at 60 °C, the absorbance of the solutions was measured at 562 nm using a Thermo Scientific Multiskan GO, and the data were obtained using SkanIt for Multiskan GO 3.2 (Thermo Fisher Scientific, Loughborough, UK).

### 2.4. Synthesis of Hydroxyapatite on Bio-Templates

Hydroxyapatite (HAp) control on bio-templates was synthesised according to Constantini et al. [17]. The following solutions were prepared: sodium dihydrogen phosphate dihydrate (NaH_2_PO_4_ 32.5% *w*/*v*), calcium chloride dihydrate (CaCl_2_·2H_2_O 53% *w*/*v*), and ammonium hydroxide (NH_4_OH 25% *w*/*v*). CaCl_2_·2H_2_O (43.4 mL) alone or containing 130 mg of pollen grains and of NaH_2_PO_4_ (26.6 mL) were added simultaneously to the reaction vessel and stirred at 540 rpm and 37 °C. The pH was increased to and maintained at 7.0 by the careful addition of the ammonium hydroxide solution, and the pH was determined using an Accumet AB150 pH meter (Fisher Scientific, Loughborough, UK). The reaction was maintained by stirring at 37 °C and pH 7 for 24 h. The precipitate obtained was washed with deionized water and separated by centrifugation at 2000 rpm for 5 min, and the process was repeated three times; the obtained product was further dialysed (Medicell cut off 12–14 KDa) for three days against water, the water was changed twice a day, and the purified product was freeze dried. A total of two grams of product from each reaction was transferred into an open crucible and heated to 1000 °C for two hours in a Carbolite Furnace (Sheffield, UK) and allowed to cool; any changes in mass were recorded [18,19]. Samples were stored in a desiccator until further use.

### 2.5. Physicochemical Characterisation

Pollen grains, before and after purification, after calcification, and after calcination were characterised using the techniques described below.

#### 2.5.1. X-ray Diffraction Analysis (XRD)

The samples were analysed at 25 °C with an X’Pert^3^ Powder X-ray Diffractometer (Malvern Panalytical, Malvern, Worcestershire, UK) equipped with a copper tube and a PSD detector. The following instrumental parameters were applied: voltage 40 kV; current 35 mA; 2θ range 5–90°; step size 0.0130°; scan step time 97.92 s; stage spinner enabled; Ni-filter for Kβ radiation; and PSD opening 3.35° (2θ). The raw files of the measurements were converted using the program Raw File Exchange (Bruker AXS, Karlsruhe, Germany). The software EVA 14.2 (Bruker AXS), combined with the PDF-2 database (Powder Diffraction File, International Centre for Diffraction Data, Newtown Square, PA, USA), was used to identify the phases and compare the X-ray patterns.

#### 2.5.2. Scanning Electron Microscopy (SEM)

The surface morphology of the samples was analysed by acquiring micrographs with a Zeiss EVO MA10 Scanning Electron Microscope (Carl Zeiss, Jena, Germany). The samples were coated with gold and palladium (Quorum Q150RS, Quorum Technologies, Puslinch, ON, Canada) and the images were captured at 15 kV. Cross-sections of the calcified pollen grains were obtained following embedding into epoxy resin/hardener (MetPrep, Coventry, UK), and polishing with 800 and 1200 grit silicon carbide papers (MetPrep, Coventry, UK). Before imaging, the samples were cleaned with ethanol in an ultrasonic bath (XUBA3, Grant Instruments, Cambridge, UK) for 15 min and were air dried. The images were acquired using TESCAN Mira3 FEG-SEM OI (Brno, Czech Republic) to gain insights into the powder structure. The images were obtained at a working distance of 12 mm, an acceleration voltage of 5 kV, and a beam current of 0.8 nA.

#### 2.5.3. Fourier-Transform Infrared Spectroscopy (FTIR) Analysis

A FTIR was performed using a Varian 600-IR series (Varian Medical Systems, Palo Alto, CA, USA). The spectra were collected in the region of 4000 to 400 cm^−1^ using a resolution of 4 cm^−1^, and a total of 16 scans were recorded for each sample.

#### 2.5.4. Thermal Gravimetric Analysis (TGA)

A TGA of the materials was performed with a Discovery 5500 (TA Instruments, New Castle, DE, USA) between 20 and 900 °C with a heating rate of 10 °C min^−1^ under a N_2_ atmosphere (50 mL min^−1^).

#### 2.5.5. Laser Diffraction

Particle size was measured by laser diffraction with a Sympatec Helos (H1885) particle size analyser using a Rodos dispenser fitted with an Aspiros doser for dry powders (feed velocity 25 mm/s, pressure 2 bar) (Sympatec, Manchester, UK). The laser filter used had a range between 0.25 and 87.5 μm and the software used for analysis was Windox 5.2.1.0.

### 2.6. Statistical Analysis

Statistical analysis was performed using a two-tailed paired *t*-test and a one-way analysis of variance test (ANOVA), followed by a Tukey’s multi-comparison test using GraphPad Prism 8.3.1 (GraphPad Software, Boston, MA, USA, www.graphpad.com, accessed on 20 February 2024). A *p* value of <0.05 was deemed to be statistically significant.

## 3. Results

### 3.1. Purification and Analysis of Sporopollenin Capsules

The pollen grains were washed to eliminate impurities, genetic material, and potentially immunogenic proteins. In order to evaluate the success of the acid washings, SEM characterisation, FTIR analysis, and protein analysis were conducted. The percentage yield of the purification process for each pollen species was calculated to determine the efficiency of the process (Table 1). Overall, the acidolysis resulted in a low yield process with a maximum yield lower than 20% for ragweed pollen and a yield as low as 3.5% for the stargazer lily pollen. The types of pollen selected for this study presented distinct and characteristic morphological features (Figure 1) [4]. Raw alder pollen grains exhibited an oblate shape with a slightly textured exine (scabrate or granulate) surface and a main visible aperture per grain (Figure 1 and Appendix A). After washing (Figure 1 and Appendix A), the surface of the exine capsule showed a more defined texture and its length was significantly shorter (Table 1, *p* = 0.0003). The unwashed dandelion pollen grains (Figure 1 and Appendix A) were characterised by a unique morphology with a triangular outline and a reticulate appearance [13]. Following washing, the dandelion pollen grains (Appendix A) exhibited burst lacunae; echini or spines of a more defined shape could be observed along the lophae or ridges of the reticulate.

A significant reduction in size (*p* = 0.0026) was observed. Raw lamb’s quarter pollen (Figure 1 and Appendix A) presented a spheroidal shape and pantoporate surface (apertures covering the entire surface). Each grain had between 20 and 65 apertures, which measured 1.8 ± 0.14 µm. The apertures were circular and equally distributed across the pollen grain. Post acid treatment (Figure 1 and Appendix A), the pollen measured 22.5 ± 1.37 µm, significantly less than before treatment (*p* < 0.0001) and could be seen to have a rougher exine layer with evident granulation. The pores stayed the same size following the treatment; however, some ruptured pores could be seen. Unwashed ragweed grains (Figure 1 and Appendix A) appeared to be spherical with a spiked exine, and each grain appeared to have one main aperture that was closed with a membrane underneath the exine. Following the acid treatment (Figure 1 and Appendix A), the spikes were more defined and the membrane closing the aperture was ruptured. Prior to washing, stargazer lily grains (Figure 1 and Appendix A) exhibited a reticulate morphology with equidimensional bochi (lumen part of the reticulum) and an overall elliptical shape. After washing (Figure 1 and Appendix A), the muri or the walls of the bochi seemed thickened and some of the grains appeared to have burst. Successful washing of the pollen grains was confirmed by FTIR (Figure 1), and in all cases, the wide peak above 3000 cm^−1^ assigned to the stretching vibration of -OH and -NH_2_ groups of biomacromolecules such as proteins, lipids, and nucleic acids was visibly reduced. Also, the intensity of the peaks in the fingerprint area was notably reduced. Protein quantification before and after washing also confirmed the successful removal of the protein content (Figure 2) [20,21]. For all pollens, less than 1.5% of the starting amount of protein was left after the washing procedure. Dandelion and lamb’s quarter grains were defatted before the washing procedure; however, this fact did not have an effect on the efficacy of protein removal.

### 3.2. Pollen Grains as Bio-Templates

Following the purification process, ragweed and lamb’s quarter pollen grains were used as templates for the synthesis of calcium phosphate salts in mild conditions; 37 °C and neutral pH [22]. These two grains have a similar size and overall shape, but the former presents a single pore, a spiky surface, and it not defatted, while the latter is pantoporate, has a smooth surface, and is defatted. Both washed and unwashed grains were used as templates and the obtained calcified materials were characterised before and after calcination. The calcification of the pollen grains was successful for both ragweed and lamb’s quarter and both starting from unwashed or washed grains. Spherical particles were observed in all cases, accompanied by other particles of various shape and size due to the formation of calcium salts that self-nucleated in the reaction mixture (Figure 3 and Figure 4). Backscatter images of cross-sectioned ragweed samples also showed the particles obtained were hollow and that the calcination process successfully degraded the sporopollenin shell (Figure 3D,H). Ragweed grains presented a mono-distributed particle size profile with an average diameter (VMD) of 21.83 μm in agreement with the SEM images (Figure 3), and the spiked nature of their exine was responsible for a lower value of Sauter median diameter (SMD) due to the higher surface area in comparison to a smooth sphere of equal volume (Figure 5). The calcified samples were less homogeneous in their distribution and presented a higher VDM value due to the formation of a calcified layer over the pollen grain exine. The marked decrease in SMD indicated a decrease in volume/surface area ratio due to the formation of an irregular coating with increased surface area. The calcination of the samples did not have a significant effect on SMD. Lamb’s quarter grains were also mono-distributed with a VMD of 26.56 μm and the pantoporate nature of the exine led to a much lower SMD value of 16.66 μm (Figure 6); the difference between VMD and SMD was more marked for lamb’s quarter than for ragweed grains. The calcification and calcination of lamb’s quarter grains led to a decrease in SMD, as already noticed for ragweed. Also, in this case, this is supported by the SEM images (Figure 4) that show the formation of a highly textured surface on the grains, responsible for the decreased volume/surface ratio of these samples. VDM increased supporting the formation of an external layer on the exine structure.

The XRD patterns of all the calcified samples clearly revealed the presence of HAp (Figure 7). The marked broadening of the peaks indicate a very fine crystal size, a typical feature of synthetic hydroxyapatite nucleated at 25–37 °C [23]. In particular, the overlap of the three peaks at 31.7°, 32.2°, and 32.8° (Figure 7) is consistent with the presence of HAp nanocrystals [17]. A careful examination of the XRD patterns highlighted small differences between the different samples. In fact, the diffractograms of ragweed (both unwashed and washed) overlap with that of the control, whereas the peaks in the patterns of the lamb’s quarter samples are sharper, thus indicating a larger crystal size, particularly in the unwashed material (Appendix A).

Calcination of the samples always determined two effects (Figure 7B–F and Appendix A): a strong increase in the intensity of HAp peaks, and the nucleation of ß-tricalcium phosphate (ß-Ca_3_(PO_4_)_2_, rhombohedral, space group *R3c*, hereinafter ß-TCP). The XRD patterns of the calcinated samples were found to be very similar to each other (Appendix A). Namely, unwashed ragweed overlapped HAp control, while the peaks of ß-TCP resulted slightly weak for unwashed lamb’s quarter. In regards to the calcinated materials, in comparison to the HAp control, both washed samples showed slightly lower intensity peaks, and ß-TCP gave stronger reflections in the washed ragweed than in the washed lamb’s quarter, whereas the opposite situation occurred for HAp peaks (Appendix A).

Calcination is known to increase the size of HAp crystals [24,25] and thermal processing is used for improving the crystallinity [25]. On the other hand, the formation of ß-TCP testifies that a Ca-deficient apatite (i.e., Ca/P molar ratio < 1.67) was produced during the calcification process, since the treatment at 1000 °C of Ca-deficient HAp leads to the formation of stoichiometric HAp (which crystallizes in the same space group *P6_3/m_*) and ß-TCP, whereas, when Ca/P = 1.67, the HAp structure withstands up to 1300 °C [26]. Generally, the peak broadening determined by a very fine crystal size does not allow all structural substitutions to be revealed by XRD [25], especially in the case of Ca-deficient HAp (Ca_10−x_(HPO_4_)_x_(PO_4_)_6−x_(OH)_2−x_ where 0 ≤ x ≤ 1), whose diffraction patterns appear identical to those of stoichiometric HAp [26].

According to the TGA data (Figure 8 and Figure 9), both ragweed and lamb’s quarter pollen grains contain a small amount of water, below 10%. Water content was reduced after washing; this is due to the loss of protein content and thus the water adsorbed to the macromolecules. After the formation of the calcium phosphate salts, the water content remained higher in the unwashed grains compared to the washed ones. Finally, the calcination process eliminated all water content from all samples. The TGA profile of ragweed grains showed a mass loss of about 61% between 100 and 400 °C and of 33% between 400 and 650 °C; these are assigned to the degradation of the intine and the exine, respectively [27]. In the washed sample, a 20% mass loss was recorded at the lower temperature range and assigned to the degradation of the intine with a 56% loss between 350 and 550 °C due to the degradation of the exine. Similarly, for lamb’s quarter, the mass reduction corresponding to the intine degradation decreased from 46% to 17% after washing of the grains in acid, while the mass loss associated with the degradation of the sporopollenin increased from 22% to 63%. Unwashed and washed samples of pollen grains contained a small amount of material that did not decompose at 900 °C (Figure 8B). After calcification, the mass remaining at the end of the experiment was 90% of the starting mass, indicating that the organic components (plus water) of the pollen grains account for around 10% of the total mass. After calcination, almost all organic material was decomposed with only about 0.2 to 0.4% mass lost.

## 4. Discussion

Pollen grains have complex structures, varied according to their function in the specific plant species, which provides a wide range of morphologies to be used as bio-templates [8]. Allergies caused by pollen are widespread in the world population [28], and for a long time, it was thought that the main causative agent of pollen-associated allergies was the protein content of the grains. We now know that other biomolecules carried by pollen grains can be responsible for the activation of the immune response, including molecules naturally associated with pollen, such as lipids, protease, and NADPH oxidase, or pollutants of gaseous or particulate nature that adsorb to the grains [29]. Harsh washing conditions, using organic solvents, strong alkali, and strong acids can be employed to eliminate allergens without affecting the main structure of the pollen grain. The structure is formed by sporopollenin, a very resilient, highly cross-linked biopolymer made of aliphatic and phenolic acids [30]. Most pollens can be cleaned through acidolysis and base hydrolysis; however, this method has been shown not to be successful in pollens such as ragweed (*Ambrosia artemisiifolia*) and sunflower (*Helianthus annuus*) due to the formation of an irreversible insoluble cake from which the pollen grains cannot be salvaged [4,5]. This limitation has led to many other methods being produced, with a combination of acidolysis, base hydrolysis, and the use of ionic liquids [14].

The washing step is employed in many works aimed at developing sporopollenin- based materials for biomedical applications. In our study, a first washing step was employed to investigate if washing influenced the further use of pollen grain as bio-templates. The short, one-step acidolysis employed in our study was found to be effective at reducing protein content more than 98.5% for all of the pollen grains used. This was evident from the protein content assay but also from the SEM images showing more defined morphologies and opening of apertures. Further confirmation of the effect of washing was provided by the TGA profiles, with a decreased mass loss at 150–300 °C, indicating significant loss of the intine [15]. However, the recovery yield was very low in all cases, and no correlation was observed between the yield and whether the pollen was defatted or not before the acidolysis. Repeated experiments would be required to confirm this. Our intended application is the use of grains as bio-templates for the synthesis of calcium phosphate salts. The final step of the procedure is the calcination at high temperatures that degrades all organic components. If we demonstrate that the washing step has no effect on the final product, we can overcome the limitation of the low yield of this process.

Ragweed and lamb’s quarter grains were selected as two spherical templates with different surface morphologies for the synthesis of HAp in mild conditions [17]. The reaction was successful, with both templates affording spherical hollow particles. The formation of the inorganic layer on the surface of the particles was evidenced by the increase in particle size, and the porosity of the surface was demonstrated by the decrease in the Sauter median diameter for both pollen templates. No trend was observed in the change in diameter or volume/surface ratio based on the use of unwashed or washed pollen grains, suggesting that the presence of surface proteins does not play a major role as a site of nucleation for the precipitation of HAp. XRD analysis confirmed that when ragweed grains were used, the calcification process was not affected by the washing procedure, and only a slight difference in the crystals size of Ca-deficient HAp was inferred in the case of unwashed (larger crystals) and washed lamb’s quarter. Furthermore, the XRD patterns of the calcined samples turned out to be very similar to each other, evidencing that the washing step had almost no effect on the final result. The compounds obtained after the thermal treatment, HAp and ß-TCP, are both calcium phosphates exploited for bone (and tooth) repair [23]. ß-TCP is one the most used and powerful synthetic bone graft substitute because it is osteoconductive and osteoinductive [31]. Moreover, a biphasic calcium/phosphate (BCP) mixture, which is an intimate association of highly crystalline HAp and β-TCP in various proportions, offers a double advantage, since the more soluble ß-TCP dissolves first and stimulates bone growth, while HAp, less soluble, acts as an immediate scaffold as well as a slow-release agent capable of providing ions for bone formation [23,32].

Future work will focus on testing if this novel biomimetic material could support enhanced bone integration and regeneration around an implant, in order to decrease the patient’s likelihood of requiring revision surgery in the future. This application is supported by the highly porous nature of the material produced, with a large surface area to encourage bone integration, facilitate drug absorption, and control release kinetics if required, as well as the biphasic composition. Since osteoinduction and osteoconduction have been shown to be dependent on the geometry, pore size, and phase-dependent solubility of calcium phosphate materials [33,34], the use of a variety of pollen grains as templates could provide a way of obtaining biomaterials with tuneable properties. These materials can be included in scaffolds for bone regeneration (e.g., hydrogels, electrospun fibres, 3D printed structures) or they can be used to create implant coatings that favour osseointegration both for musculoskeletal and dental applications [35,36]. Future translational study should also focus on determining the efficiency of the synthetic procedure with the aim of increasing yield and reducing the final cost of the product.

## Figures and Tables

**Figure 1 biomimetics-09-00159-f001:**
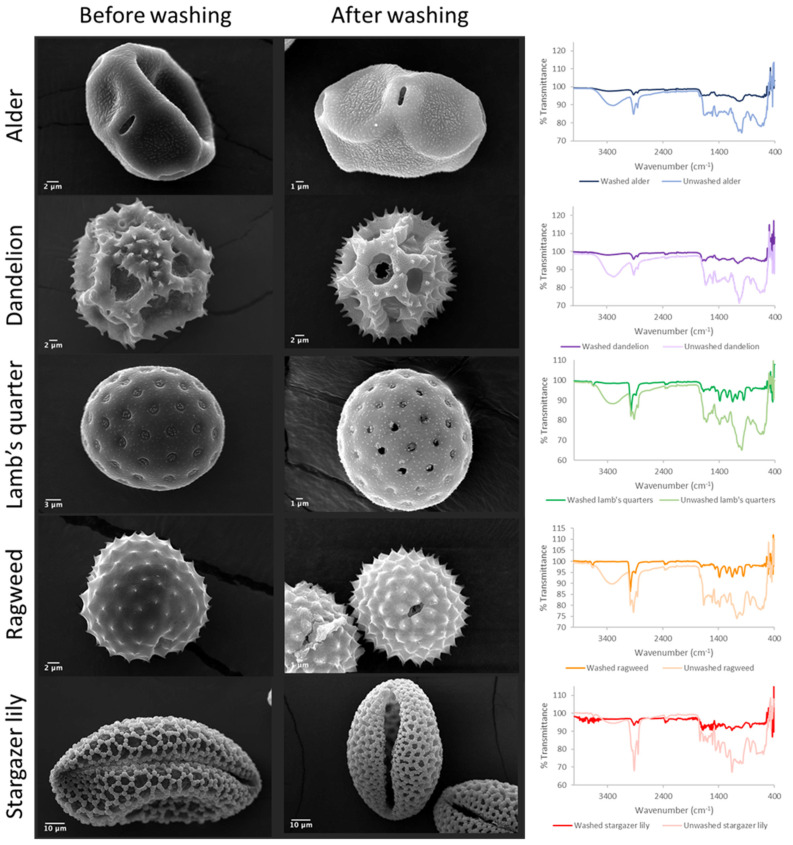
SEM images of pollen grains before and after acid washings and their respective FTIR spectra.

**Figure 2 biomimetics-09-00159-f002:**
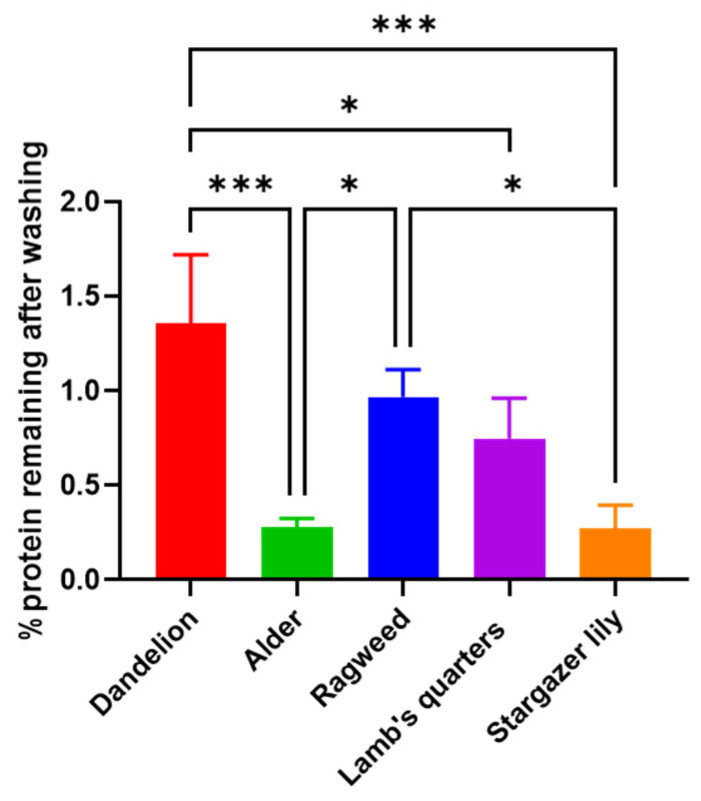
Percentage protein remaining after washing of pollen grains. Data are reported as mean ± SD (n = 3). One-way Anova returned *p* = 0.0003, Tukey’s multi-comparison post-hoc results are reported in the graph, * *p* < 0.05, *** *p* < 0.001.

**Figure 3 biomimetics-09-00159-f003:**
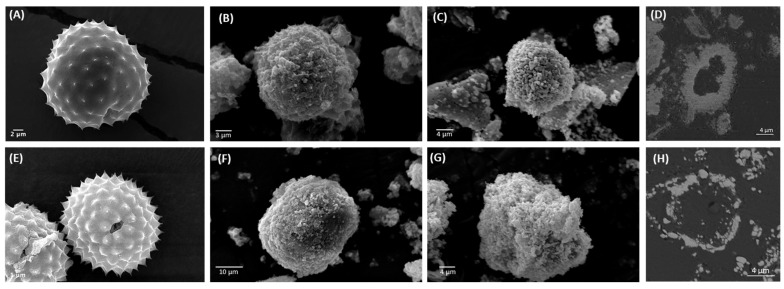
SEM images of (**A**) unwashed ragweed, calcium phosphate salts synthesised on unwashed ragweed grains (**B**) before and (**C**) after calcination; (**D**) backscatter cross-section image of unwashed ragweed before calcination; (**E**) washed ragweed, calcium phosphate salts synthesised on washed ragweed grains (**F**) before and (**G**) after calcination; and (**H**) backscatter image of unwashed ragweed after calcination.

**Figure 4 biomimetics-09-00159-f004:**
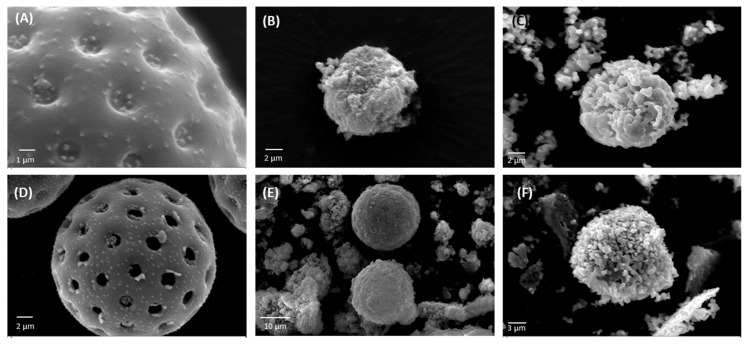
SEM images of (**A**) unwashed lamb’s quarter, calcium phosphate salts synthesised on unwashed ragweed grains (**B**) before and (**C**) after calcination; (**D**) washed lamb’s quarter, calcium phosphate salts synthesised on washed ragweed grains (**E**) before and (**F**) after calcination.

**Figure 5 biomimetics-09-00159-f005:**
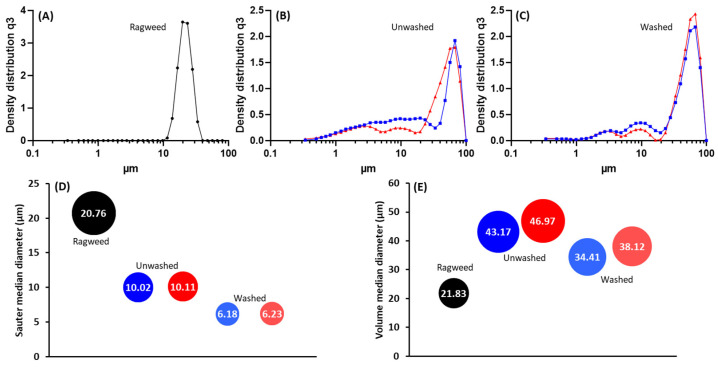
Particle size distribution for (**A**) unwashed ragweed, (**B**) unwashed ragweed after calcification (blue) and after calcination (red), (**C**) washed ragweed after calcification (blue) and after calcination (red); (**D**) sauter median diameter (SMD) and (**E**) volume median diameter (VMD).

**Figure 6 biomimetics-09-00159-f006:**
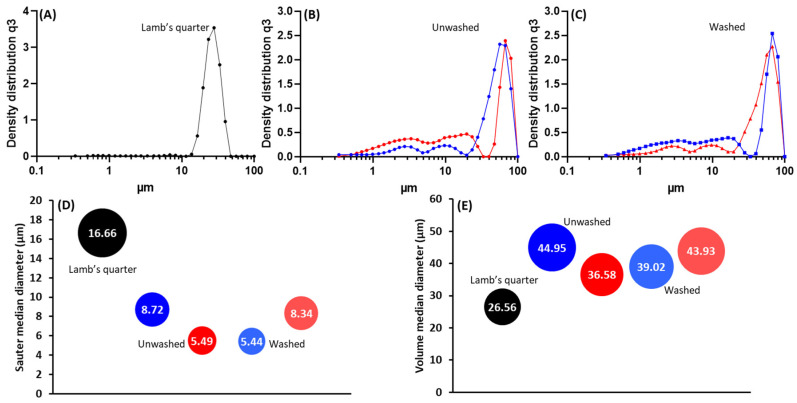
Particle size distribution for (**A**) unwashed lamb’s quarter, (**B**) unwashed lamb’s quarter after calcification (blue) and after calcination (red), (**C**) washed lamb’s quarter after calcification (blue) and after calcination (red); (**D**) sauter median diameter (SMD) and (**E**) volume median diameter (VMD).

**Figure 7 biomimetics-09-00159-f007:**
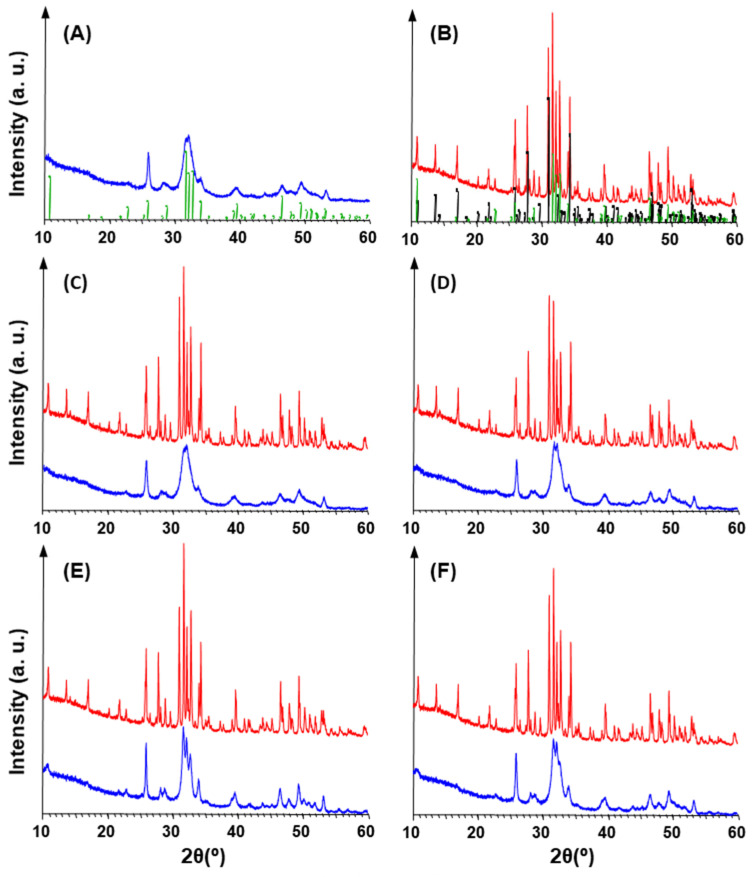
XRD patterns (in the 2θ range 10–60°) for synthesised HAp control (**A**) before and (**B**) after calcination. Diffractograms, shifted along Y-axis for clarity, of calcified (blue) and calcinated (red) samples for (**C**) unwashed ragweed, (**D**) washed ragweed, (**E**) unwashed lamb’s quarter and (**F**) washed lamb’s quarter. Green bars: HAp (PDF N. 72-1243). Black bars: β-TCP (PDF N. 70-2065).

**Figure 8 biomimetics-09-00159-f008:**
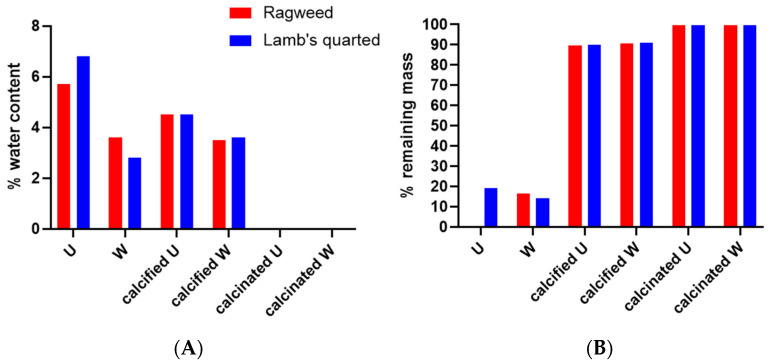
Water content (**A**) and remaining mass (**B**) for ragweed and lamb’s quarter pollen grains as calculated from TGA analysis. U stand for unwashed and W stands for washed.

**Figure 9 biomimetics-09-00159-f009:**
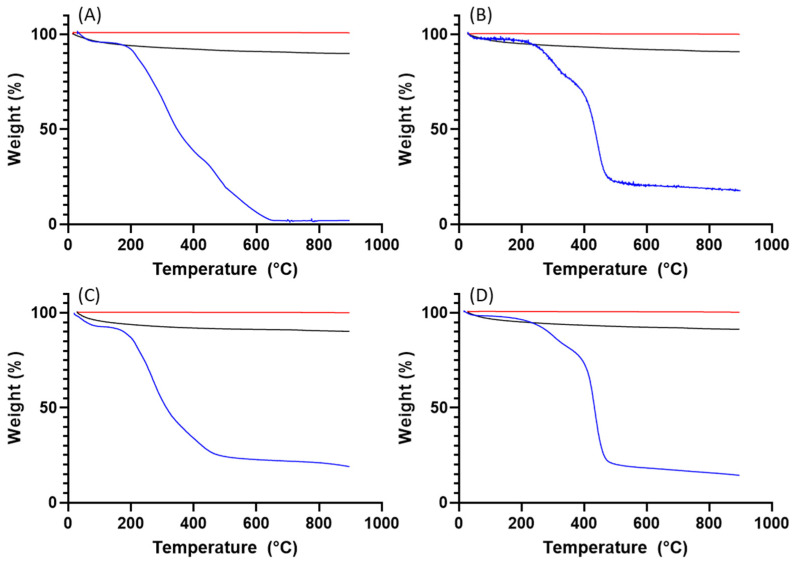
TGA analysis for (**A**) ragweed, (**B**) washed ragweed, (**C**) lamb’s quarter and (**D**) washed lamb’s quarter. Blue lines are for the pollen grains, black for the calcified pollen grains, and red for the calcinated samples.

**Table 1 biomimetics-09-00159-t001:** Percentage yield for each pollen species following acid washing (n = 1) and change in dimension (as measured from SEM images) from before to after washing. Data are reported as mean ± SD (n = 5). Data were analysed with an unpaired two-tailed *t*-test; ^a^ *p* = 0.0003; ^b^ *p* = 0.0026; ^c^ *p* < 0.0001; ^d^ *p* = 0.0174; ^e^ *p* = 0.0018.

Pollen Species	Defatted	Percentage Yield (%)	Length Unwashed (µm)	Length Washed (µm)
Alder	No	6.5	24.9 ± 0.7	21.7 ± 0.9 ^a^
Dandelion	Yes	15.9	33.6 ± 2.7	26.0 ± 2.9 ^b^
Lamb’s quarters	Yes	5.0	27.4 ± 0.6	22.5 ± 1.4 ^c^
Ragweed	No	19.8	20.7 ± 1.2	17.2 ± 2.3 ^d^
Stargazer lily	No	3.5	107.0 ± 16.5	74.7 ± 4.5 ^e^

## Data Availability

Data are available upon request to the corresponding author.

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
