# Peer review of "Sporopollenin Capsules as Biomimetic Templates for the Synthesis of Hydroxyapatite and β-TCP"

_biomimetics, 2024, doi:10.3390/biomimetics9030159_

Round 1
Reviewer 1 Report
Comments and Suggestions for Authors
This manuscript demonstrated how to use pollen grains as green and sustainable bio-templates for synthesizing biomaterials. The authors should also discuss how these templates can be used in biomedical applications.
Author Response
How the templates can be used in biomedical applications has been added at the end of the discussion:
Since osteoinduction and osteoconduction have been shown to be dependent on the geometry, pore size and phase dependent solubility of calcium phosphate materials [33,34], the use of a variety of pollen grains as templates could provide a way of obtaining bio-materials with tuneable properties. These materials can be included in scaffolds for bone regeneration (e.g. hydrogels, electrospun fibres, 3D printed structures), they can be used to create implants coatings that favour osseointegration both for musculoskeletal and dental applications [35,36].
Reviewer 2 Report
Comments and Suggestions for Authors
In my humble opinion, this interesting and well-written manuscript might be published as is. No corrections are necessary.
Author Response
The authors thank the reviewer for taking the time to assess our manuscript and for the positive comments.
Reviewer 3 Report
Comments and Suggestions for Authors
The manuscript is well planned and well written. Only minor technical comments:
- Line 93, shouldn’t be PBS?
- Part 2.4. Please provide concentrations in percent by weight or molar concentrations. % w/v concentrations are mainly used for insoluble substances.
- The abbreviation HAp should be used, not Hap.
The weakest point, however, is the lack of discussion on the economic aspect of using pollen as templates for the synthesis of calcium phosphates. What is the efficiency of this process and its costs? Moreover, won't unwashed pollen with organic residues cause allergies in practical use? I think that this part of the discussion should be added by the authors.
Author Response
The manuscript is well planned and well written. Only minor technical comments:
The authors thank the reviewer for the positive comments.
- Line 93, shouldn’t be PBS?
This has been corrected.
- Part 2.4. Please provide concentrations in percent by weight or molar concentrations. % w/v concentrations are mainly used for insoluble substances.
Concentrations provided are those used in the preparation of solutions therefore it is weight of salt per volume of solvent (% w/v)
- The abbreviation HAp should be used, not Hap.
Abbreviations have been changed in both the main manuscript and the supplementary information.
The weakest point, however, is the lack of discussion on the economic aspect of using pollen as templates for the synthesis of calcium phosphates. What is the efficiency of this process and its costs? Moreover, won't unwashed pollen with organic residues cause allergies in practical use? I think that this part of the discussion should be added by the authors.
In lines 376-377 we addressed the issue of possible allergenic residues and suggested that the calcination process would eliminate any risk as all organic material would be degraded at high temperature.
Comments on the low yields of the washing step have been made in lines 377-378.
A counterguarding the economic aspect has been added at the end of the discussion.
Future translational study should also focus on determining the efficiency of the synthetic procedure with the aim of increasing yield and reducing the final cost of the product.
Reviewer 4 Report
Comments and Suggestions for Authors
Referee report is in attached file

Author Response

(The authors gave the same response as above.)
